# Quorum Quenching Strains Isolated from the Microbiota of Sea Anemones and Holothurians Attenuate *Vibrio*
*corallilyticus* Virulence Factors and Reduce Mortality in *Artemia*
*salina*

**DOI:** 10.3390/microorganisms10030631

**Published:** 2022-03-16

**Authors:** José Carlos Reina, Pedro Pérez, Inmaculada Llamas

**Affiliations:** 1Department of Microbiology, Faculty of Pharmacy, University of Granada, 18071 Granada, Spain; jose.carlos.reina.cabello@gmail.com (J.C.R.); pedropereferre@gmail.com (P.P.); 2Biomedical Research Center (CIBM), Institute of Biotechnology, University of Granada, 18100 Granada, Spain

**Keywords:** quorum quenching, *N*-acylhomoserine lactone, acylase, marine bacteria, sea anemones, holothurians

## Abstract

Interference with quorum-sensing (QS) intercellular communication systems by the enzymatic disruption of *N*-acylhomoserine lactones (AHLs) in Gram-negative bacteria has become a promising strategy to fight bacterial infections. In this study, seven strains previously isolated from marine invertebrates and selected for their ability to degrade C6 and C10-HSL, were identified as *Acinetobacter junii*, *Ruegeria atlantica*, *Microbulbifer echini*, *Reinheimera aquimaris*, and *Pseudomonas sihuiensis*. AHL-degrading activity against a wide range of synthetic AHLs were identified by using an agar well diffusion assay and *Agrobacterium tumefaciens* NTL4 and *Chromobacterium violaceum* CV026 and VIR07 as biosensors. High-performance liquid chromatography-mass spectrometry (HPLC-MS) analysis indicated that this activity was not due to an AHL lactonase. All the strains degraded *Vibrio coralliilyticus* AHLs in coculture experiments, while some strains reduced or abolished the production of virulence factors. In vivo assays showed that strains M3-111 and M3-127 reduced this pathogen’s virulence and increased the survival rate of *Artemia salina* up to 3-fold, indicating its potential use for biotechnological purposes. To our knowledge, this is the first study to describe AHL-degrading activities in some of these marine species. These findings highlight that the microbiota associated with marine invertebrates constitute an important underexplored source of biological valuable compounds.

## 1. Introduction

In-depth research into pathogens shows evidence that virulence gene expression in many bacteria is controlled by a sophisticated cell-to-cell communication system known as quorum sensing (QS) [1]. This mechanism enables bacteria to sense their population density and to coordinate behaviors in response to changing environments. In the case of Gram-negative pathogens, they produce and detect diffusible signal molecules, mainly *N*-acylhomoserine lactones (AHLs), which accumulate in the surrounding medium as cell density increases. Once a threshold concentration of AHLs is reached, an array of phenotypes, such as exoenzymes, antibiotics and exopolysaccharides, swarming motility, and biofilm formation are regulated [2,3,4,5,6].

Traditionally, antibiotics and chemical products have been used to control bacterial diseases, although their excessive and inappropriate use has increased the emergence and spread of antimicrobial resistance (AMR) [7]. This dramatic situation constitutes a serious global threat to human and animal health which is responsible for thousands of deaths annually and is expected to increase in the coming years if alternative bacterial control strategies are not developed (CDC 2019) [8]. International organizations, such as FAO and WHO, have therefore implemented a global plan focused on the responsible use of antimicrobials and on the search for novel strategies to prevent and combat bacterial infectious diseases (www.fao.org, accessed on 17 February 2022). One promising novel ecofriendly alternative is the disruption of QS systems, whose resistance levels are expected to be lower than those observed for conventional antimicrobial treatments [9,10,11]. This includes, among others, interference with signal molecule detection by the production of quorum-sensing inhibitors (QSIs) and the enzymatic degradation of QS signal molecules, known as quorum quenching (QQ) [12]. To date, most studies have focused on AHL disruption enzymes that degrade or modify the signal molecules, which prevents the activation of QS systems. Three main groups of QQ enzymes, consisting of acylases (amide hydrolysis), lactonases (lactone hydrolysis) and oxidoreductases (oxidoreduction), have been identified [13]. The first QQ enzyme, named AiiA, was identified in a *Bacilllus* sp. strain and is expressed in the agricultural pathogen *Erwinia carotovora* that causes a decrease in the production of enzymatic activity, as well as an attenuation in its virulence [14]. Subsequently, many AHL-degrading enzymes have been identified, some of which have shown the potential to be used as biocontrol agents in aquaculture [15,16,17,18] and agriculture [19,20,21,22,23,24].

In respect of the aquaculture, bacterial diseases are mainly caused by the proliferation of opportunistic *Vibrio* species that are ubiquitous in marine and estuarine ecosystems. *Vibrio* spp. originate a high mortality rate worldwide in bivalve mollusks and are responsible of important economic losses in this sector [25]. Although *Vibrio* species, such as *V. harveyi*, *V. anguillarum*, and *V. parahaemolyticus*, are the major causes of this disease, other emerging species, such as *V. mediterranei*, *V. owensii*, and *V. corallilyticus*, are related to mortality outbreaks during the last years [25,26]. QS systems have been studied in the genus *Vibrio* and it has been demonstrated to control the expression of virulence genes, such as exoenzymes or pigment production, as well as biofilm formation (reviews: [6,27]). In the case of *V. mediterranei* VibC-Oc-097, *V owensii* VibC-Oc-106 and *V. corallilyticus* VibC-Oc-193, AHLs were identified as responsible for the production of exoenzymes and the swarming motility [28].

The marine environment is considered an excellent source of novel enzymes or compounds that interfere with QS systems [6,29,30,31,32,33]. Indeed, AHL-degrading activity has been described in many marine bacteria belonging to different genera, such as *Alteromomas*, *Bacillus*, *Paracoccus*, *Pseudoalteromonas*, *Psychrobacter*, *Roseovarius*, *Ruegeria*, *Stenotrophomonas*, *Tenacibaculum*, and *Thalassomonas* [6,30,34,35,36], as well as to marine metagenomic collections [37,38,39].

In an attempt to explore the production of novel active biological compounds, marine symbiotic bacteria from the microbiota of sea anemones and holothurians were previously screened for the production of antimicrobial molecules [40], as well as QSI compounds and QQ enzymes [35,41]. In this study, the AHL-degrading capacity of seven of these previously selected strains was tested against a wide range of synthetic AHLs, and their enzymatic mechanisms were analyzed by HPLC-MS. The ability of QQ strains to interfere with the QS system of the pathogen *V. coralliilyticus* VibC-Oc-193 was also evaluated in coculture experiments by in vitro and in vivo assays using the model *Artemia salina*.

## 2. Materials and Methods

### 2.1. Bacterial Strains, Media, Compounds, and Culture Conditions

*Acinetobacter junii* M1-66, M2-33 and M2-40; *Ruegeria atlantica* M3-98; *Microbulbifer echini* M3-111; *Rheinheimera aquimaris* M3-127 and *Pseudomonas sihuiensis* M4-84 were isolated from the microbiota of sea anemones and holothurians. The marine invertebrates were obtained from the iMARE Natural S.L. aquaculture facility (http://www.imarenatural.com, accessed on 17 February 2022) located in Motril, Granada, in the south of Spain (36°44′33.4″ N 3°31′12.1″ W) [40]. These marine isolates, which were selected in a previous study based on their capacity to degrade AHLs (35), were routinely grown at 28 °C in marine broth (MB, Difco^®^, Franklin Lakes, NJ, USA). The aquaculture-related pathogens *Vibrio coralliilyticus* VibC-Oc-193, *V. mediterranei* VibC-Oc-097 and *V. owensii* VibC-Oc-106 [42] were grown in MB at 28 °C.

The biosensor strains for AHL detection were cultured as follows: *Agrobacterium tumefaciens* NTL4 (pZLR4) [43] at 28 °C in Luria Bertani (LB) medium (10 g tryptone, 5 g yeast extract and 10 g NaCl per Liter) supplemented with 2.5 mmol L^−1^ CaCl_2_ and 2.5 mmol L^−1^ MgSO_4_ (LB/MC), or AB medium (3 g K_2_HPO_4_, 1 g Na_2_H_2_PO_4_, 1 g NH_4_Cl, 0.3 g MgSO_4_·7H_2_O, 0.15 g KCl, 0.01 g CaCl_2_, 0.0025 g FeSO_4_·7H_2_O, and 5 g glucose per liter) containing 50 µg gentamycin mL^−1^. *Chromobacterium violaceum* CV026 [44] and *C. violaceum* VIR07 [45] were grown at 28 °C in LB medium supplemented with kanamycin (50 µg mL^−1^).

The synthetic AHLs used were C4-HSL (*N*-butyryl-DL-homoserine lactone), C6-HSL (*N*-hexanoyl-DL-homoserine lactone), 3-O-C6-HSL (*N*-3-oxo-hexanoyl-DL-homoserine lactone), C8-HSL (*N*-octanoyl-DL-homoserine lactone), 3-O-C8-HSL (*N*-3-oxo-octanoyl-DL-homoserine lactone), C10-HSL (*N*-decanoyl-DL-homoserine lactone), 3-OH-C10-HSL (*N*-3-hydroxydecanoyl-DL-homoserine lactone), C12-HSL (*N*-dodecanoyl-DL-homoserine-lactone), and 3-O-C12-HSL (*N*-3-oxo-dodecanoyl-DL-homoserinelactone) (Sigma^®^, St. Louis, MO, USA).

### 2.2. Synthetic AHL QQ Activity

The QQ activity of the marine strains was assessed against a wide range of synthetic AHLs using an agar well diffusion assay [17,46]. Briefly, each synthetic AHL was added to 500 µL of a 24 h culture of each bacterial strain at a final concentration of 10 µM and incubated at 28 °C. As negative control, the same concentration of each AHL was added to 500 µL of cell-free MB medium and processed under similar conditions. The remaining AHLs in each case were detected by loading 100-µL aliquots of cell-free supernatant in wells made on the surface of LB or AB medium plates, supplemented with 80 µg of 5-bromo-4-chloro-3-indolyl-ß-d-galactopyranoside (X-Gal) per mL, in which an overlay of the biosensor strains *C. violaceum* CV026, *C. violaceum* VIR07, and *A. tumefaciens* NTL4 (pZLR4) had been previously placed, respectively. The plates were incubated at 28 °C for 24 h while awaiting the development of purple or blue haloes, respectively, around the wells, which reveals the presence of AHLs. This assay was repeated three times.

### 2.3. Identification of the Type and Location of QQ Activity

To determine whether the QQ activity of marine bacterial isolates was due to a lactonase mechanism, a lactone ring closure assay was carried out [47]. Briefly, C10-HSL (10 µM) was added to 1000 µL of an overnight culture and incubated at 28 °C for 24 h. As a negative control, cell-free MB medium was supplemented with the same AHL concentration. An aliquot of each culture (500 µL) was centrifuged and HCl 1N was added to the supernatant until reaching pH 2. Then, they were incubated for 24 h at 28 °C. The remaining AHLs, with or prior acidification, were extracted twice with a volume of dichloromethane. Dried extracts were resuspended in methanol 70 % (*v*/*v*) or 500 µL of acetonitrile and analyzed by using the well diffusion agar-plate method in LB plates with an overlay of *C. violaceum* VIR07 and high-performance liquid chromatography-mass spectrometry (HPLC-MS), respectively [48]. This assay was repeated three times.

To identify the localization of the QQ enzyme of each marine bacterial isolate, the crude cellular extract (CCE) and supernatant fractions were obtained and QQ activity in each case was determined by the agar well diffusion method as described elsewhere [49]. To obtain CCE, bacterial cells from overnight cultures were resuspended in PBS buffer (pH 6.5), sonicated and then filtered through a 0.22 µm-pore membrane filter. The supernatant of the cultures was also filtered through a 0.22 µm-pore membrane filter.

### 2.4. In Vitro Coculture Assays

Cocultures assays of each marine bacterium and *Vibrio* spp. were performed according to a methodology described elsewhere [17,35]. Briefly, the pathogens (10^7^ CFU mL^−1^) were cocultured with each bacterial isolate (10^9^ CFU mL^−1^) at a ratio of 1:100 in MB medium and incubated for 24 h at 28 °C. As controls, each bacterium was monocultured under similar conditions. The remaining AHLs from each coculture and monoculture were quantified using an agar well diffusion assay as described above, with *A. tumefaciens* NTL4 (pZLR4) as biosensor strain. The abundance of each bacterium was determined in the cocultures by serial dilutions and plate counts using Marine Agar (MA) and Thiosulfate Citrate Bile Sucrose (TCBS) media.

To evaluate the interference with cellular functions controlled by QS in *V. coralliilyticus* VibC-Oc-193 by bacterial isolates with QQ activity in cocultures, a diverse range of enzymatic activities were tested. Thus, 10 µL of the cocultures and monocultures were spotted on different media. Proteolytic and hemolytic activity was determined in casein medium and in blood agar medium, respectively [50]. Swimming and swarming motility was measured in soft agar (MB 0.3% and 0.5% agar (*w*/*v*), respectively) [51,52]. In the motility assays, only 2 µL of the cocultures were spotted in the center of the plates. Alkaline phosphatase activity was determined in MA medium supplemented with phenolphthalein phosphate 0.01% (*w*/*v*). Chitinase activity was detected in MB supplemented with colloidal chitin [53]. Amylase activity was determined in MA supplemented with starch 1% (*w*/*v*). Gelatinase and DNAase activities were detected with gelatin (Difco^®^, Franklin Lakes, NJ, USA) and DNAase (Cultimed^®^, Madrid, Spain) medium, respectively. These assays were repeated three times.

### 2.5. In Vivo Assays against Vibrio coralliilyticus

The ability of the bacterial isolates to interfere with the virulence of *V. coralliilyticus* VibC-Oc-193 was assessed in vivo in *Artemia salina* (brine shrimp) nauplii according to a protocol described elsewhere [28,41]. Briefly, the hatching cysts of *A. salina* (JBL Artemio Pur) were obtained according to the manufacturer’s instructions using sterile filtered and autoclaved seawater (SFSW) (salinity 36 g L^−1^, 20 °C, pH 7.3), and groups of 20 nauplii were transferred to Petri dishes containing 20 mL of SFSW. Cocultures of each bacterial isolate and the pathogen were prepared in a ratio of 1:100 and then added. As controls, monocultures of each bacterium (QQ strains and pathogen) and the same volume of SFSW were used. In each treatment, the bacterial cells were washed with SFSW, added to the nauplii (10^6^ CFU mL^−1^) and incubated at 25 °C for 3 days. Three replicates of each treatment were performed, and the experiment was repeated three times. The survival of the shrimps was scored daily after the addition of bacteria. The growth of *V. coralliilyticus* VibC-Oc-193 in monoculture and coculture was tested by the plate counting method using the selective TCBS medium. 

### 2.6. Statistical Analysis

The results were analyzed using the software R with an ANOVA test followed by a Tukey *t* test. Differences at *p* < 0.05 were considered statistically significant.

## 3. Results

### 3.1. Characterization of QQ Activity of AHL-Degrading Bacteria

The seven strains tested in this paper were selected in a previous study on the basis of their high capacity to degrade C6 and C10-HSL [35]. They were isolated from the microbiota of invertebrate marine species, such as sea anemones and holothurians [40] and taxonomically identified by sequencing and analyzing their partial 16S RNA gene sequences as follows: *A. junii* (strains M1-66, M2-33, M2-40); *R. atlantica* M3-98; *M. echini* M3-111; *R. aquimaris* M3-127; and *P. sihuiensis* M4-84 [35]. 

In this study, we evaluated the capacity of the seven strains to degrade a wide range of unsubstituted, oxo- and hydroxyl-substituted synthetic AHLs (C4-HSL, C6-HSL, 3-oxo-C6-HSL, C8-HSL, 3-oxo-C8-HSL, 3-OH-C10-HSL, C10-HSL, C12-HSL, and 3-oxo-C12-HSL) using an agar well diffusion assay. The biosensor strains used were *C. violaceum* CV026 to detect C4-HSL, C6-HSL and 3-oxo-C6-HSL; *C. violaceum* VIR07 to detect 3-OH-C10-HSL and C10-HSL and A. tumefaciens NTL4 (pZLR4) to detect the medium- and long-chain AHLs. The presence of exogenous AHLs was observed in relation to the development of purple and blue colors, respectively. The assay was repeated three times the halo diameter was measured in each case and compare with the negative control (Appendix A). As shown in Table 1, the strains degraded a broad range of AHL with high levels of activity observed, being strongest against long-chain AHLs. In the case of C6-HSL, 3-O-C6-HSL, C8-HSL, and 3-O-C8-HSL, some of the tested strains showed partial or no activity, although all of them degraded C4-HSL. 

### 3.2. Identification and Localization of QQ Enzymes

An acidification assay was performed in order to determine whether the QQ activity of the seven strains was due to a lactonase-type enzyme. Therefore, supernatants from 24 h culture of each bacterium were acidified to pH 2 after incubation with C10-HSL. If the QQ activity was due to lactonase, the open lactone ring would be recirculated under acidic conditions and the AHL concentration recovered. A parallel assay was performed for each strain at pH 7. As a negative control, a similar concentration of AHL was added to MB and tested under similar conditions (pH 2 and pH 7). AHL extracts from each sample were qualitatively tested by an agar well diffusion assay (Figure 1a) and quantitatively analyzed by HPLC-MS (Figure 1b). The results in all cases indicated that the AHL concentration was not restored under acidic conditions. Thus, the QQ activity of the seven AHL-degrading strains was confirmed not to be due to a lactonase mechanism.

To determine the cellular localization of the QQ enzymes, supernatant and CCE from each bacterium were obtained and tested for AHL-degrading activity against C10-HSL. In all cases, QQ activity was detected in CCE. 

### 3.3. Interference with Vibrio spp. QS Systems and Impact on Associated Phenotypes Using Coculture Experiments with AHL-Degrading Strains

Based on the results above, we assessed whether the AHL-degrading strains had the capacity to degrade AHLs produced by pathogenic aquaculture-related species in order to evaluate their potential in vivo use. We, therefore, tested the QQ activity of the seven strains by coculturing them with three *Vibrio* pathogens: *V. coralliilyticus* VibC-Oc-193, *V. mediterranei* VibC-Oc-097 and *V. owensii* VibC-Oc-106. In these pathogens, the production of AHLs have been characterized by high performance liquid chromatography-Fourier transform-high resolution mass spectrometry (HPLC/FT-HRMS) and correlated to the expression of virulence factors. The main AHLs produced were C4-HSL and 3-OH-C10 in *V. coralliilyticus* VibC-Oc-193; 3-O-C12-HSL and C4-HSL in *V. mediterranei* VibC-Oc-097 and C12-HSL and 3-OH-C12-HSL in *V. owensii* VibC-Oc-106 [28]. Prior to the coculture experiments, an antagonism assay was performed to confirm that each AHL-degrading strain did not interfere with the growth of *Vibrio* spp. The results indicated that there was no negative effect on the growth of the pathogen strains in any cases (Appendix A).

The three *Vibrio* spp. were cocultured with each of the AHL-degrading strains in MB medium at a ratio of 1:100. Monocultures of each bacterium were used as controls. After 24 h of incubation at 28 °C, the remaining AHLs were detected by an agar well diffusion assay using the biosensor *A. tumefaciens* NTL4 (pZLR4). As mentioned above, the presence of exogenous AHLs was observed according to the development of blue color. In general, the concentration of AHLs in the cocultures was decreased as compared to the control performed with the pathogen monoculture. The results indicated that no AHLs were produced by the seven QQ strains. AHLs produced by *V. coralliilyticus* VibC-Oc-193 and *V. owensii* VibC-Oc-106 were totally degraded by the seven strains, while only five strains, M1-66; M2-33; M2-40; M3-98 and M4-84, were able to partially degrade the AHLs produced by *V. mediterranei* VibC-Oc-097 (Table 2; Appendix A).

Coculture assays of the seven AHL-degrading strains and the pathogenic *V. coralliilyticus* VibC-Oc-193 were conducted under conditions similar to those described above. This *Vibrio* species was selected for the coculture experiments due to its virulence and the high QQ capacity showed by the strains tested to degrade its AHLs. After 24 h incubation, the effect on the production of cellular functions, including virulence factors regulated by QS in this pathogen strain, was analyzed. Some phenotypes of *V. coralliilyticus* VibC-Oc-193 were affected when cocultured with the AHL-degrading strains (Figure 2; Appendix A). The production of amylase in this pathogen was reduced when cocultured with strains M1-66 and M2-33 and abolished when cocultured with M4-48. The production of hemolysin was reduced in the presence of M2-33, M3-98 and M3-127 and abolished when cocultured with strains M1-66 and M4-84. The swimming and swarming motility of the pathogen was decreased in the presence of M3-111. The production of DNAase was slightly reduced when cocultured with M2-40, M3-98 and M4-84, and the production of V. *coralliilyticus* VibC-Oc-193 alkaline phosphatase was slightly reduced when cocultured with M4-84, while caseinase and lipase production (Tween 20 hydrolysis) was unaffected in all cases. 

### 3.4. Interference with Vibrio coralliilyticus VibC-Oc-193 Virulence in Artemia salina by AHL-Degrading Strains

The potential use of AHL-degrading strains to control bacterial infections was evaluated by testing their QQ activity in a virulence assay in the aquaculture *A. salina* model. Cocultures of *V. coralliilyticus* VibC-Oc-193 and each bacterium were prepared in a 1:100 ratio and monocultures of each bacterium were used as controls. The results indicated that, in the majority of cases, the virulence of the pathogen in *A. salina* nauplii was reduced when cocultured with all the strains as compared to the *V. coralliilyticus* VibC-Oc-193 monoculture, with this reduction being statistically significant after 48 and 72 h of infection (Figure 3). The survival rate of *Artemia salina* infected by the pathogen increased up to 2-fold in the presence of M2-40 and M4-84 and 3-fold in the presence of strains M3-111 and M3-127 (Figure 3).

To rule out a growth inhibition of *V. coralliilyticus* VibC-Oc-193 by the AHL-degrading strains, a plate counting method was used. No differences in the growth of the pathogen were observed in the monocultures with respect to the cocultures (10^7^ CFU mL^−1^).

## 4. Discussion

The widespread use of antibiotics to control bacterial infections in aquaculture has induced an increase in antibiotic resistance mechanisms. The search for novel therapeutic targets and new approaches to ensure the effectiveness of treatments against multiresistant pathogens is, therefore, one of the main goals of current research by international organizations such as FAO and WHO (www.fao.org, accessed on 17 February 2022). QS disruption, which has become a promising strategy to combat bacterial infections [54,55], appears to induce less selective pressure, thus reducing the potential emergence of resistance [56,57].

One QS interruption mechanism, the enzymatic degradation of AHLs (QQ), has been demonstrated to have the potential for use against bacterial infections in aquaculture. For example, different QQ strains or QQ enzymes have been reported to increase the survival rate of fish and crustaceans infected by *Aeromonas hydrophila* [16,18] or *Vibrio* spp. [17,28,58,59].

Marine environments are considered an underexplored source of molecules and natural products with biological activity [60,61,62,63]. In fact, it is estimated that 91% of species inhabiting oceans have not yet been described, and the associated microbiota of marine invertebrate are an important and promising source of novel active substances [63,64]. On this basis, we previously carried out an extensive search for QS-inhibiting compounds among bacteria isolated from the microbiota of sea anemones and holothurians [35]. As a result, seven strains, identified as *A. junii* (strains M1-66, M2-33, and M2-40), *R. atlantica* M3-98; *M. echini* M3-111; *R. aquimaris* M3-127 and *P. sihuiensis* M4-84, with a high capacity to degrade C6 and C10-HSL, were isolated and selected for further study.

In this study, the seven strains selected showed QQ activity against a broad spectrum of synthetic AHLs molecules, with the long-chain AHLs observed to be more efficiently degraded. This finding is in line with the enzymatic activity of the majority of AHL-degrading bacteria previously described [35,36,48,65]. To determine whether AHL-degrading enzyme activity was due to a lactonase, samples of the seven bacteria were incubated in the presence of C10-HSL and subsequently acidified to enable the lactone ring to be recirculated. No significant recovery in the AHL tested was observed in a well diffusion agar assay in any cases. This was corroborated by HPLC-MS analysis, suggesting that the enzymatic activity was not caused by a lactonase. It has been reported that QQ acylase enzymes in cultivable marine bacteria [17,35,36,48,66,67] and in marine metagenomic collections [37,38] seem to be more abundant than lactonases. 

The species *A. junii* is widely recognized as a potential pathogen, although it is rarely capable of causing infections [68,69]. The use of this species in biotechnology has also been demonstrated. For example, in biomedicine, the antioxidant properties of the lipopeptide produced by a strain of *A. junii* and its ability to enhance wound healing have been demonstrated [70]. *A junii*, with its capacity to degrade phosphates, has also been used in water treatment plants [71] and has even been proposed as a bioremediation tool thanks to its ability to accumulate lead [72]. Nevertheless, to our knowledge, this study is the first to describe QQ activity in *A. junii*. The existence of lactonase-type and acylase-type QQ enzymes in other species of the same genus has been reported in other studies [23,66,72,73,74].

Importantly, to our knowledge, this study is the first to describe the noteworthy QQ activity of the strain *R. aquimaris* M3-127, a marine gammaproteobacterium, against a wide variety of AHLs. Curiously, a strain of *R. aquimaris* capable of interfering with QS systems through the production of a diketopiperazine has been reported [75]. Recently, it has also been described that synthetic diketopiperazines modified from the natural compound that is present in this species is capable of interfering with QS-related phenotypes in *P. aeruginosa* [76].

In addition, as far as we know, although QQ enzyme production has not been described with respect to *R. atlantica* M3-98, a marine alphaproteobacterium, a lactonase-type QQ enzyme has been identified in a *R. mobilis* strain [77]. However, our assays have proven that the QQ activity of the strain M3-98 was not due to a lactonase, with our study being the first to find a non-lactonase enzyme in this genus. 

We also demonstrated the presence of QQ activity in *M. echini* M3-111, a marine gammaproteobacterium. Though little is known about the genus *Microbulbifer*, in recent years, multiple compounds with biotechnological potential have been identified in this genus. For instance, different species of this genus can produce antibacterial compounds [78] or enzymes of industrial interest, such as β-glucosidases and alginate-lyases [79,80,81]. In this study, we describe for the first time the presence of a non-lactonase QQ enzyme in this genus, thus broadening its biotechnological potential.

Lastly, another QQ strain identified in this study is *P. sihuiensis* M4-84. The presence of QQ enzymes, such as the acylases PvdQ, QuiP, HacA, and HacB, in the genus *Pseudomonas* has frequently been described, all of which are found in *P. aeruginosa* [82,83,84]. Recently, QQ activity has also been identified in *P. segetis* P6, in which a non-lactonase QQ enzyme has been reported to be responsible for degrading AHLs and to act as a biocontrol tool for agriculture pathogens [22].

In order to evaluate the potential use of the QQ bacteria selected to control bacterial infections in aquaculture, the capacity to degrade AHLs produced by three *Vibrio* pathogenic species was tested by carrying out coculture experiments. Vibriosis is one of the most common pathologies to affect aquaculture facilities [25,85], and the QS systems in members of this genus have been demonstrated to control different virulence mechanisms [27,28,86].

Coculture assays revealed that all the QQ strains were able to degrade the AHLs produced by the pathogenic species *V. coralliilyticus* VibC-Oc-193, *V. mediterranei* VibC-Oc-097 and *V. owensii* VibC-Oc-106, and no inhibitory effect on pathogen growth was shown. The impact of AHL degradation on the production of virulence factors was analyzed in the aquaculture pathogen *V. coralliilyticus* VibC-Oc-193 by performing coculture experiments. Since some QQ strains also produce hydrolytic enzymes or motility, results regarding to some cellular activities in the pathogen could not be proven. Thus, the production of amylase, hemolysin, and alkaline phosphatase in the pathogen was abolished in the coculture with the strain M4-84. These enzymatic activities were partially inhibited in the cocultures with the strains M1-66, M2-33, M3-98, M3-127, while swarming and swimming motilities were reduced in the presence of strain M3-111.

A reduction in the production of phenotypic characteristics related to the virulence of the pathogen *V. coralliilyticus* VibC-Oc-193 was confirmed, with the QQ strains being observed to increase the survival rate of *A. salina* nauplii. This well-known model of infection in aquaculture has previously been used to assess the virulence of different *Vibrio* species, as well as to check the reduction in their virulence when QS disruptors are used [28,87,88]. A remarkable reduction in the virulence of the pathogen was observed in the presence of the strains *M. echini* M3-111 and *R. aquimaris* M3-127, resulting in up to a 3-fold increase in the survival rate of *A. salina*.

Previous studies with other strains isolated from the microbiota of anemones and holothurians (*S. maltophilia* and *Psychrobacter* sp.) had shown similar results, these bacteria being capable of increasing the survival rate of *A. salina nauplii* [35,36]. Nevertheless, the 3-fold increase seen in this work is higher than the data obtained in the mentioned studies. The use of QQ bacteria as biocontrol in aquaculture has also been proposed to other species, such as *Bacillus* spp. [89,90] that were able to increase the survival rate of fish and crustacean models. Nevertheless, up to our knowledge, this is the first report of *A. junii*, *M. echini*, *R. aquimaris*, and *P. sihuiensis* as biocontrol strains in aquaculture. Other strains of those genera with QQ activity have shown potential as biocontrol tools. For example, *A. calcoaceticus* reduced *Fusarium oxysporum* infection in tomato [91], and *Acinetobacter* sp. XN-10 reduced the pathogenicity of *P. carotovorum* subsp. *carotovovum* in carrots, potatoes and Chinese cabbage [23]. *Pseudomonas* QQ-strains have also been recently used as biocontrol tools in agriculture [22,24], but this report shows its potential also in aquaculture. Overall, these results demonstrate the potential effectiveness of using AHL degradation in the fight against vibriosis in aquaculture, as well as marine invertebrate microbiota as a source of bacteria with QQ activity.

## 5. Conclusions

In conclusion, in this study, AHL-degrading activity caused by a non-lactonase enzyme is described for the first time in the marine species *A. junii*, *M. echini*, *R. aquimaris* and *P. sihuiensis*. Some of these QQ strains reduce or inhibit hydrolytic activities, as well as swimming and swarming motilities in the pathogen *V. coralliilyticus* VibC-Oc-193, as evidenced by coculture experiments. The strains *M. echini* M3-111 and *R. aquimaris* M3-127 significantly attenuated *V. coralliilyticus* VibC-Oc-193 infection in *A. salina*. Our results indicate that strains M3-111 and M3-127 could be promising candidates for use as biocontrol agents in aquaculture and a potential alternative to antibiotics against vibriosis.

## Figures and Tables

**Figure 1 microorganisms-10-00631-f001:**
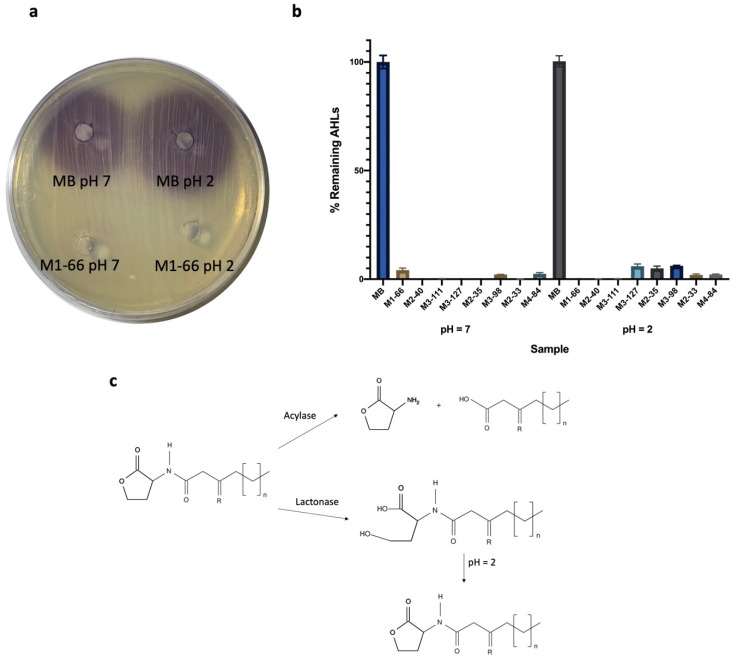
Detection of the remaining C10-HSL after incubation with the AHL-degrading strains at pH 7 and pH 2. (**a**) Diffusion agar-plate assay to detect C10-HSL using the biosensor strain *C. violaceum* VIR07. (**b**) HPLC-MS measurements of C10-HSL. Values are referred to as % remaining AHLs. Cell-free MB medium was used as negative control. Initial AHL concentration was 10 µM. This assay was repeated three times. Error bars show standard deviations. (**c**) Schematic description of the QQ reactions that may occur.

**Figure 2 microorganisms-10-00631-f002:**
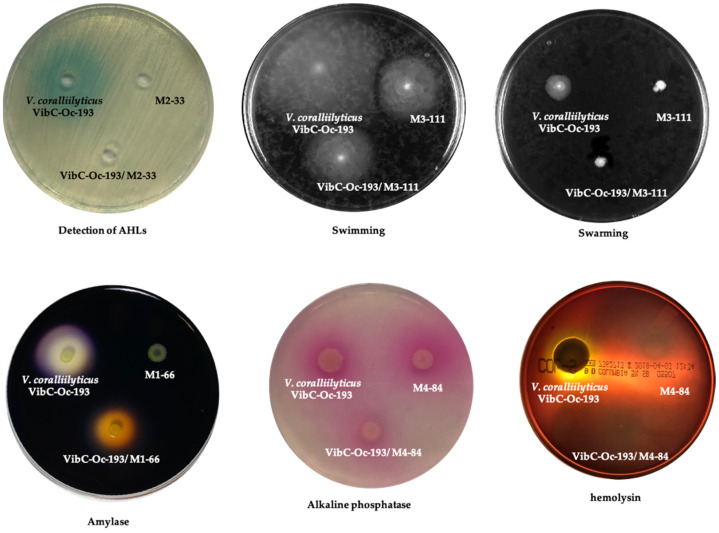
Detection of AHLs and phenotypes (enzymatic activities and motility) in the monocultures and co-cultures of AHL-degrading strains and the pathogen *V. corallilyticus* VibC-Oc-193. *A. tumefaciens* NTL4 (pZLR4) was used as biosensor strain to detect AHLs.

**Figure 3 microorganisms-10-00631-f003:**
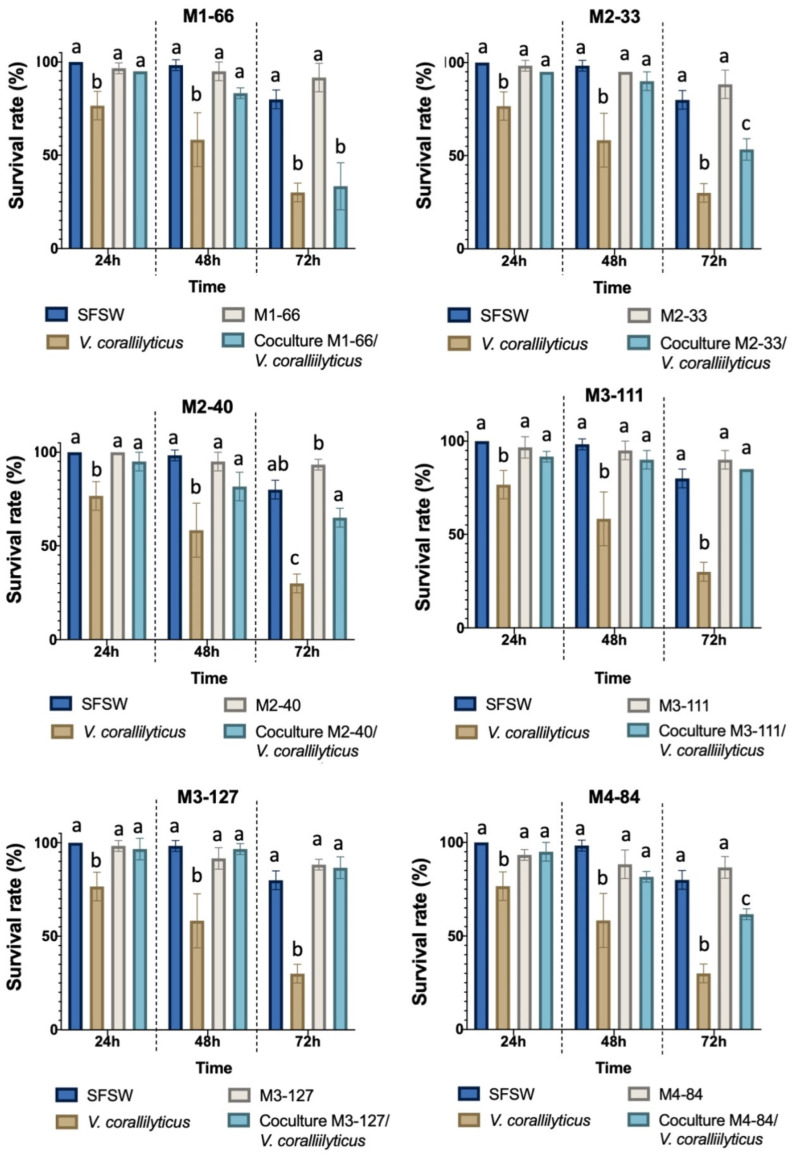
Survival rate of *A. salina* nauplii after 24 h, 48 h and 72 h of incubation with *V*. *coralliilyticus* VibC-Oc-193 in the presence or absence of AHL-degrading strains. Sterile filtered seawater (SFSW) was used as control. Different letters above the bars indicate that the values are significantly different (*p* < 0.05). The statistical analysis was carried through a Tukey’s Multiple Comparison Test.

**Table 1 microorganisms-10-00631-t001:** Quorum quenching activity of marine strains against synthetic AHLs.

Strain	C4-HSL	C6-HSL	3-O-C6-HSL	C8-HSL	3-O-C8-HSL	C10-HSL	3-OH-C10-HSL	C12-HSL	3-O-C12-HSL
** *Acinetobacter* ** ***junii* M1-66**	++	++	−	++	+	+	++	++	++
** *A.* ** ***junii* M2-33**	++	++	−	++	−	++	++	++	++
** *A.* ** ***junii* M2-40**	++	++	−	++	++	+	++	++	++
** *Ruegeria* ** ***atlantica* M3-98**	++	++	+	−	++	+	++	++	++
** *Microbulbifer* ** ***echini* M3-111**	++	−	−	−	++	++	++	++	++
** *Rheinheimera* ** ***aquimaris* M3-127**	++	++	−	++	++	++	++	++	++
** *Pseudomonas* ** ***sihuiensis* M4-84**	++	++	−	++	++	+	++	++	++

Detection of QQ activity based on the activation of the biosensor strains by means of measure of halo diameter (“++” full AHL degradation; “+” partial AHL degradation: “−” no AHL degradation).

**Table 2 microorganisms-10-00631-t002:** Quorum quenching activity of marine strains against *Vibrio* spp. in coculture.

Strain	*V. coralliilyticus* VibC-Oc-19	*V. mediterranei* VibC-Oc-097	*V. owensii* VibC-Oc-106
** *Acinetobacter* ** ***junii* M1-66**	++	+	++
** *A.* ** ***junii* M2-33**	++	+	++
** *A.* ** ***junii* M2-40**	++	+	++
** *Ruegeria* ** ***atlantica* M3-98**	++	+	++
** *Microbulbifer* ** ***echini* M3-111**	++	−	++
** *Rheinheimera* ** ***aquimaris* M3-127**	++	−	++
** *Pseudomonas* ** ***sihuiensis* M4-84**	++	−	++

Detection of QQ activity based on the activation of the biosensor *A. tumefaciens* NTL4 (pZLR4) by means of measure of halo diameter (“++” full AHL degradation; “+” partial AHL degradation: “−” no AHL degradation).

## Data Availability

The data presented in this study are fully available in the main text and Appendix A of this article.

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
