# Peer review of "Quorum Quenching Strains Isolated from the Microbiota of Sea Anemones and Holothurians Attenuate *Vibrio"

_microorganisms, 2022, doi:10.3390/microorganisms10030631_

Round 1

Reviewer 1 Report

Manuscript Title: Quorum quenching strains isolated from the microbiota of sea anemones and holothurians attenuate Vibrio corallilyticus virulence factors and reduce mortality in Artemia salina

Manuscript ID: microorganisms-1614438

General comment:

In the manuscript, the authors evaluated the AHL-degrading activity of isolated quorum quenching strains against a wide range of synthetic AHLs. These strains degraded Vibrio coralliilyticus VibC-Oc-193 AHLs in coculture experiments in vitro, while some strains reduced or abolished the production of potential virulence factors,  indicating their potential use for biotechnological purpose. 

In general, this paper is clearly laid out, well planed and easy to read. The experiments are well designed and appropriate controls are presented. Some specific suggestions or questions are listed below:

  1. Abstract: use full name for HPLC-MS for the first time.
  2. Introduction: Add references to support the statement “Traditionally, antibiotics and chemical products have been used to control bacterial  diseases, although their excessive and inappropriate use has increased the emergence and spread of antimicrobial resistance (AMR).”
  3. Introduction is easy to read but needs a little completed. For example,there are many AHL quorum quenching strains ioslated from various environments in recent years, see the publication (doi:10.1128/AEM.02065-19; doi:10.3389/fmicb.2021.694161; doi:10.3390/microorganisms8050697; doi:3390/microorganisms8081100; doi:10.3389/fmicb.2020.00898), I suggest the authors include more information into this section and cite the recent research into the field.
  4. The Introductionsection should focus on the research progress related to the topic and emphasize the innovation of this research. However, the novelty and significance of the topic were not highlighted, please modify the introduction more clearly.
  5. I suggest the authors add a new section in the Materials and Methods to describe the details of the statistical analysis.
  6. Results : Please include “*Detection of QQ activity based on the activation of the biosensor Agrobacterium tumefaciens NTL4 (pZLR4)”as Figure 1.
  7. Discussion: why the authorsdid not compare the biocontrol efficacy of isolated quorum quenching strains in this study with that of other AHL-degrading strains based on the literature? This way the authors will demonstrate that they really have a good knowledge of the related literature. I strongly suggest the authors add more discussion into this section and cite the recent research into the field.
  8. Conclusions: “Acinetobacter junii, Microbulbifer echini, Reinheimera aquimaris and Pseudomonas sihuensis”, Abbreviations and acronyms should be defined the first time they are used within the main text and then used throughout the remainder of the manuscript. Please check it throughout the manuscript.
  9. Manyof the references have been superceded and more modern ones are required, such as Mikrobiologiya. 1948, 362-370; 1997, 143, 3703–3711. https://doi.org/10.1099/00221287-143-12-370.;  Proc.  Natl.  Acad.  Sci.  USA.  1997,  94,  6036–6041. https://doi.org/10.1073/pnas.94.12.6036; 2002,  70,  5635-46. https://doi.org/10.1128/IAI.70.10.5635-5646.2002; J. Clin. Microbiol. 2002, 40, 2696-2697.

Author Response

We thank Reviewer 1 for his/her obvious time and effort taken with our manuscript and we truly appreciate his/her input about our paper and his/her kind opinion about it. We followed his/her constructive comments to improve our paper.

We have attached a pdf file with the corresponding answers.

Reviewer 2 Report

The manuscript titled “Quorum quenching strains isolated from the microbiota of sea anemones and holothurians attenuate Vibrio corallilyticus virulence factors and reduce mortality in Artemia salina” is a well written manuscript. Materials and Methods section is well detailed. There are few minor comments that need to be addressed:  

  1. Manuscript will be much enriched with the structures of the chemicals involved.
  2. Figure 1: the standard deviations are missing. Authors also need to include in the figure legend the numbers are replicates of how many measurements.
  3. The biochemical reactions involved should be included as a figure.
  4. Authors have mentioned about negative controls, but did they use any positive control?
  5. Figure 1 and Figure 3: It will be a good idea to include the specific statistical analysis used for that particular data set.

Author Response

We thank the reviewer for the obvious time and effort he/she has taken with our paper. His/Her comments are really appreciated and we esteem his/her input about our manuscript.

We have attached a pdf file with the corresponding answers.

Reviewer 3 Report

Dear authors,

In the present study “Quorum quenching strains isolated from the microbiota of sea anemones and holothurians attenuate Vibrio corallilyticus virulence factors and reduce mortality in Artemia salina” by Reina et al., the authors present quorum quenching (QQ) activities of seven bacterial strains previously isolated from marine hosts. The authors evaluated the AHL-degrading activities of the strains against various synthetic AHLs using the agar well diffusion assay method with Chromobacterium violaceum and Agrobacterium tumefaciens reporter strains. The authors were able to exclude a lactonase mechanism for all identified QQ activities. Living strains were tested for their AHL-degrading potential of the pathogen Vibrio coralliilyticus with the corresponding reduction of selected virulence factors. Consequently, the authors showed increased survival rates of Artemia salina by the hypothesized QQ activity.

The manuscript focuses on the identification of QQ activities as an alternative to traditional antibiotics. In an era of emerging antibiotic resistance, searching for novel combating strategies is an urgent and hot topic. In the marine system, as also the author state, it is expected that many novel antimicrobials can be identified. The present manuscript shows first interesting approaches but represents a rather basic study. The abstract appropriately summarizes the study; however, I recommend including the used methods to already allow the reader to rate the presented results. The introduction more or less covers all necessary points to understand the following results and methods. It would be desirable to get more information on the opportunistic pathogen Vibrio coralliilyticus and its Quorum sensing (QS) system and downstream regulation of virulence factors. The results and discussion section are clearly presented, but I am not convinced by the statements and conclusions.

Major concerns are:

  • the QQ mechanism was not identified, only a lactonase activity was excluded, QQ activities are simply relying on reporter assays
  • simple, fast, and cheap methods could be used to identify the AHL-QQ mechanism
  • QQ activities were identified in crude cell extract but not in culture supernatant which leaves to the open question of how can those QQ activities act on external AHLs
  • experiments on pathogen virulence reduction were conducted with living cultures, what about other effects than QQ
  • two, for my understanding, important controls were mentioned but data was not shown

Minor subjects are:

  • Figure and Table legends should be improved because they are so far not self-explaining
  • abbreviations should be introduced
  • bacteria genus/species names should be written in italics

Comments, recommendations, and questions are implemented in the attached pdf file.

Author Response

(The authors gave the same response as above.)

Round 2

Reviewer 1 Report

The authors have considered all comments raised by the reviewers and revised the manuscript accordingly based on these comments. The revision is fine and can be accepted for publication.

Just last comments: Please correct "P.s sihuiensis" as "P. sihuiensis" in the Conclusions section. Please use full name for the genus name for the first time  in the Table 1 and Table 2,  and delete the wavy line.

Reviewer 3 Report

In the present study “Quorum quenching strains isolated from the microbiota of sea anemones and holothurians attenuate Vibrio corallilyticus virulence factors and reduce mortality in Artemia salina” by Reina et al., the authors present quorum quenching (QQ) activities of seven bacterial strains previously isolated from marine hosts. 
The manuscript focuses on the identification of QQ activities as an alternative to traditional antibiotics. In an era of emerging antibiotic resistance, searching for novel combating strategies is an urgent and hot topic. In the marine system, many novel antimicrobials can be identified. 

The authors improved the manuscript by including the reviewer´s recommendations. Figures were improved and legends edited. The abstract now appropriately summarizes the study. The introduction was optimized by adding more information and references and now covers all necessary points to understand the following results and methods. The results and discussion section are clearly presented and were significantly improved by adding missing information and reviewer comments. The point-by-point response clarified all open questions.